# Cryo-EM structure of the RC-LH core complex from an early branching photosynthetic prokaryote

Yueyong Xin [1], Yang Shi [2,3], Tongxin Niu[2,3], Qingqiang Wang[1], Wanqiang Niu[1], Xiaojun Huang[4], Wei Ding[4], Lei Yang[1], Robert E. Blankenship [5], Xiaoling Xu [1] & Fei Sun [2,3,4]

Photosynthetic prokaryotes evolved diverse light-harvesting (LH) antennas to absorb sunlight and transfer energy to reaction centers (RC). The filamentous anoxygenic phototrophs (FAPs) are important early branching photosynthetic bacteria in understanding the origin and evolution of photosynthesis. How their photosynthetic machinery assembles for efficient energy transfer is yet to be elucidated. Here, we report the 4.1 Å structure of photosynthetic core complex from *Roseiflexus castenholzii* by cryo-electron microscopy. The RC–LH complex has a tetra-heme cytochrome *c* bound RC encompassed by an elliptical LH ring that is assembled from 15 LHαβ subunits. An N-terminal transmembrane helix of cytochrome *c* inserts into the LH ring, not only yielding a tightly bound cytochrome *c* for rapid electron transfer, but also opening a slit in the LH ring, which is further flanked by a transmembrane helix from a newly discovered subunit X. These structural features suggest an unusual quinone exchange model of prokaryotic photosynthetic machinery.

[1] Hangzhou Normal University, 2318 Yuhangtang Road, Cangqian, Yuhang District, Hangzhou 311121 Zhejiang Province, China. [2] National Laboratory of Biomacromolecules, CAS Center for Excellence in Biomacromolecules, Institute of Biophysics, Chinese Academy of Sciences, 15 Datun Road, 100101 Beijing, China. [3] University of Chinese Academy of Sciences, 19 Yuquan Road, 100049 Beijing, China. [4] Center for Biological Imaging, Institute of Biophysics, Chinese Academy of Sciences, 15 Datun Road, 100101 Beijing, China. [5] Departments of Biology and Chemistry, Washington University in St. Louis, St. Louis, MO 63130, USA. These authors contributed equally: Yueyong Xin, Yang Shi, Tongxin Niu. Correspondence and requests for materials should be addressed to X.X. (email: xuxl@hznu.edu.cn) or to F.S. (email: feisun@ibp.ac.cn)

Photosynthesis is the primary solar energy transformation process that powers life on Earth[1]. In order to efficiently capture solar radiation, diverse photosynthetic apparatuses have evolved in different types of photosynthetic organisms. Two photosystems work together in tandem in higher plants[2,3], whereas in cyanobacteria there are supramolecular mega-complexes found[4]. Anoxygenic photosynthetic prokaryotes are diverse groups of bacteria thriving worldwide since the early history of the planet[5]. In most species of photosynthetic prokaryotes, light energy is initially absorbed by the peripheral light-harvesting (LH) antenna and then transferred via the inner LH to the reaction center (RC), where the primary reaction of photosynthesis occurs. The primary separated electron is transferred within the RC to a quinone. The fully reduced quinol is then exchanged with an oxidized quinone from the membrane pool and passes its electrons to the next redox component in the cyclic electron transfer pathway, during which a transmembrane proton gradient is established for the subsequent production of ATP[6].

In purple bacteria, the core RC–LH1 and the peripheral LH2 were found and they share a similar modular architecture[7]. The basic unit of both LHs is an αβ-heterodimer, which binds BChl *a* and carotenoid as light-harvesting pigments. In the near-infrared region, the LH2 complex has two absorption bands around 800 and 850 nm, whereas the core RC–LH1 complex has a single strong peak at 880 nm[8]. In contrast to LH2, where highly resolved X-ray structures are available[9,10], diverse structural characteristics of the RC–LH1 core complex were discovered, which are related to how quinol is released from the RC, passes through the palisade of the LH antenna and is exchanged with quinone from the membrane pool. In the crystal structure of the purple sulfur bacterium *Thermochromatium tepidum* (*T. tepidum*) RC–LH1 (called *tt* RC–LH1 hereafter, accession code 3WMM), although it assembles into a closed elliptical LH ring, varied spaces between inter-LHαβ heterodimers were observed with a maximum of 2 Å between the α-helices on the periplasmic side and 3.5 Å on the cytoplasmic side[11]. Therefore, it was proposed that the quinol/quinone can diffuse and exchange via the channels between the LHαβ heterodimers in the core complex of this bacterium[11,12].

However, in the dimeric *Rhodobacter sphaeroides* (*R. sphaeroides*) RC–LH1 (called *rs*RC–LH1 hereafter, accession code 4JC9 and 4V9G)[13,14] and the monomeric *Rhodopseudomonas palustris* (*Rhodops. palustris*) RC–LH1 (called *rp*RC–LH1 hereafter, accession code 1PYH)[15], one LHαβ heterodimer in the LH1 ring is replaced by a protein called Puf X or W, which results in a physical gap at the LH1 ring, allowing an efficient quinol/quinone diffusion. Other studies also suggested that the LH1 ring from *R. sphaeroides* may not be fully closed and contains less than 16 αβ heterodimers[16]. In addition, the existence of the gap at the LH1 ring was further confirmed by single-molecule spectroscopy at low temperature with the occurrence of a narrow peak at the red wing of the spectra[17,18]. The presence of a gap has severe consequences for the electronic structure of the complex and dynamic parameters of the LH antenna excitation states[19]. Nonetheless, there is still an urgent need to get more structural evidence to understand the structural diversity of the core complex as well as the diverse quinol/quinone diffusion mechanisms.

Photosynthetic members of phylum *Chloroflexi*, the filamentous anoxygenic phototrophs (FAPs), are phylogenetically distant from other anoxygenic photosynthetic bacteria and form the deepest branch of bacteria[20]. This type of organism acquired a "chimeric" photosynthetic system during evolution, with their pheophytin/quinone reaction center resembling that in purple bacteria, while the light-harvesting apparatus is similar to green sulfur bacteria[21]. Therefore, FAPs have received considerable attention as an important group when exploring the early evolution and diversity of photosynthesis[22].

The recently described *Roseiflexus castenholzii* (*R. castenholzii*) provides an opportunity to investigate the core photosynthetic components of FAPs[23]. The photosynthetic system of *R. castenholzii* exhibits a similar excitation energy transfer process and trapping rate with the core complex of purple bacteria[15,23], but spectroscopically resembles a peripheral LH2 of purple bacteria with the absorption peaks at 804 and 880 nm[23,24], which suggested that topologically it is composed of one mosaic LH antenna that is tightly bound with the RC[25,26]. The RC–LH core complex of *R. castenholzii* has been isolated and biochemically characterized[25,27]. It is composed of five subunits: the β and α subunits of the LH antenna, and the L, M, and cytochrome (Cyt) *c* subunits of the RC, which are encoded by *puf* genes in the order of *puf B, A, LM*, and *C*[23]. The L and M subunits are encoded by a fused gene *puf LM*[23] but by two independent genes in both purple bacteria and the representative FAP *Chloroflexus aurantiacus* (*C. aurantiacus*)[28]. Each L and M subunit binds three BChl and three bacteriopheophytin (BPheo)[26], instead of four BChl and two BPheo as in purple bacteria[29]. The RC of *R. castenholzii* is compositionally the smallest one among anoxygenic photosynthetic bacteria, due to the lack of the H subunit that is typically found in purple bacteria[13,23,30]. In addition, only one type of quinone (menaquinone-11) was found in the RC–LH core complex of *R. castenholzii*[22], instead of a menaquinone and a ubiquinone found in many purple bacteria[29]. These novel biochemical features make the core complex from *R. castenholzii* a key system for structural analysis, to further explore the photosynthetic mechanism in prokaryotic systems, and to understand the evolution of these systems.

The overall pigment organization of the *R. castenholzii* core complex has been characterized intensively in our previous spectroscopy studies[24–27,31]. Negative stain electron microscopy revealed the core complex from *R. castenholzii* has a similar size and shape with that of purple bacteria[11,13,15,32], and has 15 ± 1 LHαβ subunits assembled into a slightly elliptical LH ring, surrounding a tetra-heme cytochrome *c* bound to the RC[26]. Recently, an electron microscopic 3D reconstruction of the core complex with a resolution of 14.6 Å showed that the LH antenna embraces the RC to form a complete elliptical ring, with the cytochrome subunit protruding to the periplasmic space[33]. However, due to the limited resolution, molecular details about the subunit arrangement and pigment organization are still elusive.

In this work, we determined the structure of the *R. castenholzii* core complex (called *rc*RC–LH hereafter) at 4.1 Å resolution by the single-particle cryo-EM approach. The overall structure exhibits an elliptical shape with the tightly bound Cyt *c* protruding into the periplasmic space. All α, β, L, M, and Cyt *c* subunits, and the light-harvesting pigments as well as the electron transfer prosthetic groups within the complex have been clearly resolved. The cryo-EM structure reveals a physical gap of the elliptical LH ring, a distinctive transmembrane helix from Cyt *c* subunit inserts into the gap, and a newly discovered subunit X with its flexible transmembrane helix flanking the gap, suggesting an unusual quinone shuttling channel found in phototrophs. Our structure provides a framework for further investigation of the early branching prokaryotic photosystem.

## Results

**Overall structure of *rc*RC–LH complex.** In this study, we isolated and purified the intact *rc*RC–LH complex from photoheterotrophically grown *R. castenholzii* cells (Supplementary Fig. 1A, B and Supplementary Fig. 8). Each subunit of the *rc*RC–LH complex was verified by peptide mass fingerprinting

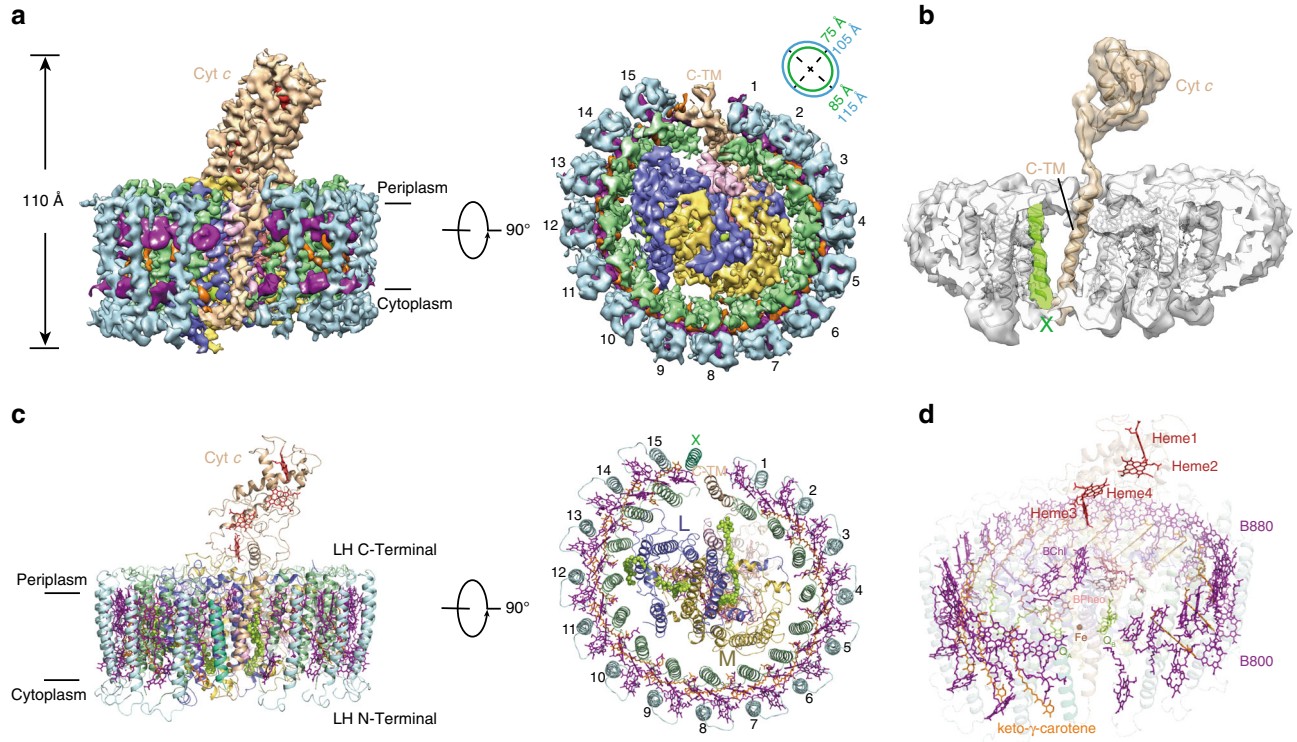

**Fig. 1** Overall structure of RC–LH complex from *R. castenholzii*. **a** The cryo-EM map of the RC–LH complex is shown at side (left) or bottom-up (right) view. The dimensions of the RC–LH complex and LH ring are represented. **b** A low-pass (6 Å) filtered cryo-EM map of the RC–LH complex, with the Cyt *c* subunit and the subunit X highlighted in wheat and green. **c** Cartoon representation of the RC–LH complex. All the cofactors are shown as sticks except the menaquinone-11 and iron molecule, which are shown as spheres. They are presenting in the same direction as **a**. **d** Stick diagram showing arrangement of the cofactors in RC–LH complex in a tilted view. Color codes for all panels: pale green, α-apoproteins; pale cyan, β-apoproteins; wheat, Cyt *c*; slate, L; yellow orange, M; light pink, the TM7 of the reaction center; green, subunit X; purple, BChls; orange, keto-γ-carotene; tv-red, heme; salmon, BPheos; limon, quinones; brown, iron

(PMF) analysis via matrix-assisted laser desorption/ionization time of flight (MALDI-TOF) and nano-flow liquid chromatography linear trap quadrupole (LTQ)-Orbitrap mass spectrometry (Supplementary Tables 1 and 2). Besides α, β, L, M, and the Cyt *c* subunits, a hypothetical subunit X (WP_041331144.1) containing 63 residues was identified (Supplementary Fig. 1A and Supplementary Tables 1 and 2).

The vitrified homogeneous *rc*RC–LH complexes were imaged on a 300 kV FEI Titan Krios cryo-electron microscope using a FEI Falcon IIIEC camera at counting mode (Supplementary Fig. 2A–B). Using the single-particle analysis approach with image classification (Supplementary Fig. 2C) and 3D refinement (Supplementary Fig. 3), the structure of the *rc*RC–LH complex was determined at an overall 4.1 Å resolution according to the gold standard FSC (Fourier shell correlation) at 0.143 (Supplementary Fig. 2D–F and Supplementary Table 3), which benefited from the highly improved signal-to-noise ratio by the Falcon IIIEC camera (counting mode) compared to the Falcon II camera (integration mode) (Supplementary Fig. 4). The final reconstructed cryo-EM map was clearly resolved for building an accurate model of the transmembrane (TM) helices with side chains, and all the other secondary structural elements as well as all the pigments (Supplementary Fig. 5 and Supplementary Movie 1).

The whole *rc*RC–LH complex is 110 Å high; its transmembrane region is composed of an elliptical LH ring (averaged diameter of ~100 Å) and the TM helices of the L, M, and Cyt *c* subunits (Fig. 1a). The soluble region of the complex is composed of the heme-binding domain of the Cyt *c* subunit that protrudes into the periplasmic space. There are 15 LHαβ heterodimers clearly

resolved from the density map, leaving a gap at one end of the elliptical LH ring. To our surprise, we observed a piece of weak density around the position of the gap (Supplementary Fig. 6). When we performed a low-pass filter of the density map to 6 Å resolution, this piece of density can be clearly resolved and modeled with a flexible TM helix (Fig. 1b, see also Supplementary Movie 2). No side chains can be assigned at this resolution. Coincidently, the above-identified hypothetical subunit X was found comprising mainly a TM helix (TMHMM Server, http://www.cbs.dtu.dk/services/TMHMM/). Therefore, hypothetically, we assigned this flexible TM helix as the part of this subunit X and designated as TMx. Besides the 15 LHαβ heterodimers as well as the TMx, L, and M subunits and the membrane-bound Cyt *c* subunit are well modeled into the density map (Fig. 1c), there are 48 BChls, 3 BPheos, 14 keto-γ-carotenes, 2 menaquinone-11, 1 non-heme iron atom, and 4 hemes clearly resolved in the density map (Fig. 1d and Supplementary Fig. 5).

**The elliptical LH ring**. The LH ring is assembled by the TM helices of 15 heterodimers of inner LHα and outer LHβ subunits, with their N terminus directed to the cytoplasmic side and C terminus on the periplasmic side. The gap of the LH ring is flanked by the transmembrane helix TMx (Fig. 1c). These observations perfectly match the previous result that 15 ± 1 subunits are composed in the antenna by high performance liquid chromatography (HPLC)[26]. The lengths of the major and minor axes are 115 Å and 105 Å for the outer ring and 85 Å and 75 Å for the inner ring (Fig. 1a). Each αβ-heterodimer contains two B880 at the periplasmic side, one B800 on the cytoplasmic side, and one

keto-γ-carotene spanning the transmembrane region (Fig. 1c, d and Supplementary Fig. 5G), which is consistent with the spectroscopic studies that $rc$RC–LH showed two absorption peaks (804 and 880 nm) in the near-infrared region[24]. The B880 pigments bound in the same and adjacent LHαβ are parallelly overlapped with each other to form the inner layer of the pigment ring, the B800 pigments are inserted between adjacent LHβ in the outer layer and are perpendicular to the B880 ring (Fig. 1c, d). We observed that there is one carotenoid molecule occupying the inter-helical space of one LHαβ, and it was built as keto-γ-carotene at the present resolution[27]. Its polyene chain is inclined to the membrane plane and connects the space between LHα at the periplasmic side and B800 of adjacent LHβ (Fig. 1c, d). The edge-to-edge distances between B880 and keto-γ-carotene in each LHαβ range from 3.5 to 4.8 Å and that between B800 and keto-γ-carotene from 2.9 to 4.2 Å (Supplementary Table 4). The edge-to-edge and center-to-center distances among adjacent B880 molecules are all within 14 Å (Supplementary Table 5), which is adequate for efficient electron orbital coupling and energy resonance[34].

**The reaction center**. In purple bacteria, L and M subunits are encoded by two independent $puf$ genes, each contains five transmembrane helices. However, the L and M subunits of $R.$ $castenholzii$ are encoded by a fused $puf$ $LM$ gene, and no typical stop codon of the tentative $puf$ $L$ was found[23]. Biochemical analysis clearly shows that the L and M subunits are separate peptides in the mature complex, so that some sort of posttranscriptional or more likely posttranslational processing must take place[23,25,26]. Hydrophobicity analysis (TMHMM Server v.2.0, http://www.cbs.dtu.dk/services/TMHMM/) of the $puf$ $LM$ gene product suggests the presence of 11 membrane-spanning regions starting from Phe67. However, 12 transmembrane helices in the reaction center were clearly resolved according to the current density map (Fig. 2a and Supplementary Fig. 5A–D). There are six transmembrane helices (TM1-6) of the L subunit and five transmembrane helices (TM8-12) of the M subunit modeled with continuous density, respectively, and an additional transmembrane helix TM7 was built with a separated density (Fig. 2a, b). The side chains of those transmembrane helices were assigned based on the high-quality density map and further verified with the conserved pigment-binding motifs.

Interestingly, the transmembrane helix TM1 that was not predicted from the hydrophobicity analysis was assigned from Lue8 to Ala31 based on the density map (Supplementary Fig. 5B). N-terminal sequencing indicated that the L subunit starts from Ser2-Ala-Val-Pro-Arg6 (Supplementary Fig. 1C, see also ref. [23]), which verifies the assignment of TM1.

The transmembrane helix TM7 is separated from the L and M subunits with its terminus 43 Å away from the C terminus of the L subunit (Gln310 in the current model) and also 43 Å away from the N terminus of the M subunit (Ile336 in the current model). N-terminal sequencing indicated that the M subunit starts from Gly327-Arg-Gly-Arg-Glu331 or Gly329-Arg-Glu-Thr-Pro333 (Supplementary Fig. 1D, see also ref. [23]). Thus, if TM7 comes from the gene product of $puf$ LM, the only available sequence to be assigned for TM7 starts from Gly311 to Leu326 and there are not enough residues from the sequence to link the long distances between the L/M subunit and TM7. Thus, TM7 could be either an unidentified new gene product or a proteolytic fragment of the $puf$ LM gene product that is cleaved twice during processing (Fig. 2b).

The L and M subunits, together, accommodate a photoreactive special pair of BChls (B865), one accessory BChl (B818) and three bacteriopheophytin (BPheo) pigments (Fig. 1d), instead of four BChl and two BPheo in purple bacteria[29]. In the sequence alignment of the LM polyprotein from $R.$ $castenholzii$ with $C.$ $aurantiacus$ and $T.$ $tepidum$, an isoleucine residue (Ile505) is found in $R.$ $castenholzii$ in place of the histidine residue that serves as a ligand to the Mg atom of the accessory BChl in M subunits of purple bacteria (Supplementary Fig. 7A, see also ref. [23]). The special pair BChls are parallel to each other and coordinated by His212 and His525 (Fig. 2c and Supplementary Fig. 7A). The overlapped B880s in the LH ring and special pair BChls are approximately located in the same plane with the nearest edge-to-edge distances of 32.6 Å, which determines the energy transfer rate from LH to RC with the decay constant 60 ps and also avoids quenching of LH pigments by the oxidized special pair[24].

Instead of a menaquinone and a ubiquinone as found in many purple bacteria[29], two menaquinone-11 ($Q_A$ and $Q_B$) were resolved in the quinone-binding pockets of the L and M subunits at the cytoplasmic side (Fig. 2c), respectively, according to the density map (Supplementary Fig. 5I) as well as previous biochemical studies[22]. $Q_A$ is buried in the intra-helical region of TM1-4 of L subunit and TM11, TM12 of M subunit. Its 1,4-naphthoquinone group is directed to a non-heme iron with a distance of 6.4 Å, and its hydrophobic tail extends to the periplasmic side of the membrane (Fig. 2c). The iron ion is coordinated by His229 at TM5, His264 at TM6, His542 at TM11, Glu557 and His 589 at TM12 (Fig. 2c and Supplementary Fig. 7A).

**Cytochrome $c$ subunit**. The Cyt $c$ subunit of $R.$ $castenholzii$ was co-purified with the L and M subunits. This tight association is special among FAPs[6,23] and also different from many purple bacteria[29]. Indeed, an unusual N-terminal transmembrane helix C-TM was resolved in the cryo-EM density map, which anchors the Cyt $c$ subunit into the membrane (Figs. 1b and 2d). Hydrophobicity analysis of the Cyt $c$ subunit suggested the existence of one transmembrane helix from Phe20 to Ile42 (TMHMM Server v.2.0, http://www.cbs.dtu.dk/services/TMHMM/). N-terminal sequencing identified the first five residues of the Cyt $c$ subunit as Gln4-Pro-Pro-Thr-Leu8 (Supplementary Fig. 1E). Guided by this information, C-TM was assigned from Val23 to Ile46 (Fig. 2e and Supplementary Fig. 5E–F). Our investigation reveals that the Cyt $c$ subunit of $R.$ $castenholzii$ binds to the RC tightly by utilizing a fusion C-TM that also interacts with the LH ring (Fig. 1a, b).

The tight association of Cyt $c$ with the $rc$RC–LH endows the capability for rapid electron donation from hemes to the photo-oxidized special pair BChls[23]. Superimposition of Cyt $c$ subunit from $rc$RC–LH with that of $tt$RC–LH1 showed that, except for the N-terminal region, the five α helices (H1–H5) that coordinate four heme molecules on the periplasmic side can be overlaid very well (Fig. 2f). Furthermore, the porphyrin rings of the four heme molecules are almost overlaid, respectively. Each iron ion in the heme molecule is bound to a His residue with the binding motif of C–X–X–C–H and a Met residue, which is strictly conserved in different species (Supplementary Fig. 7B), with the exception that both axial ligands of heme 4 in $tt$RC–LH1 Cyt $c$ are two His residues (Fig. 2g).

**LHαβ heterodimer**. In each LHαβ heterodimer of $R.$ $castenholzii$, α-B880 is coordinated by α-His27, and β-B880 is immobilized by β-His44 (Fig. 3a). These residues are conserved among FAPs and purple bacteria (Supplementary Fig. 7C and 7D). The bacteriochlorin ring of B800 is coordinated by β-His26 and β-Trp14 at one side, and the retinyl group of keto-γ-carotene from neighboring LHβ at the opposite side (Fig. 3a).

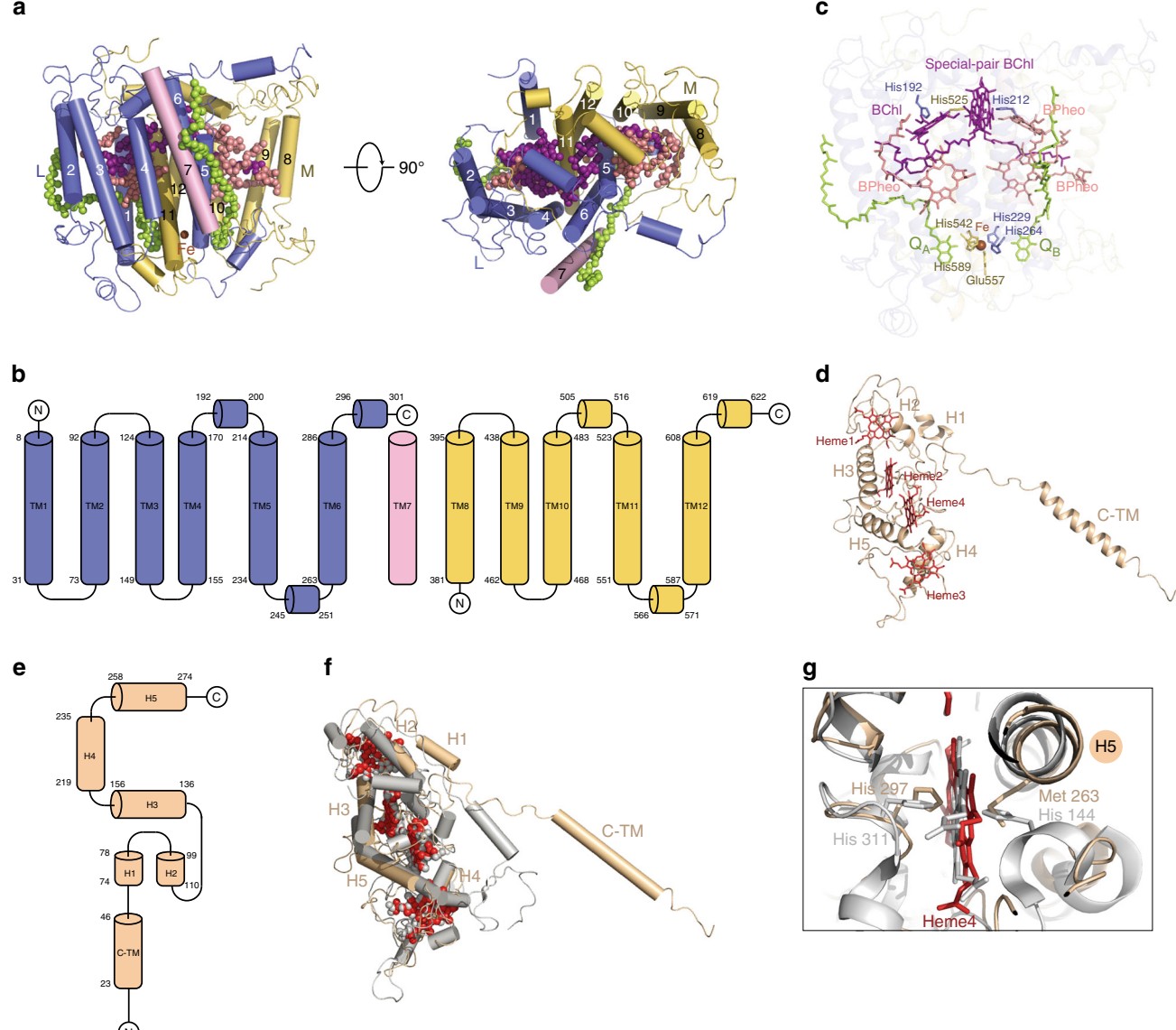

**Fig. 2** Architecture of the reaction center. **a** The cartoon presentation of the L and M subunits in side view (left) and top view (right), and the cofactors are shown as spheres. **b** The topology diagram of L and M subunits. The TM7 (light pink) is an independent transmembrane helix in the current complex. **c** The cofactors in L and M subunits. To highlight the cofactors, the apoprotein of L- and M subunits are shown as 70% transparency. The amino acids coordinate the BChl, and iron ion are shown in sticks and labeled. **d**, **e** The cartoon (**d**) and topology (**e**) diagram of the Cyt *c* subunit, the hemes are shown as red sticks. **f** Structural comparison of the Cyt *c* subunit from *T. tepidum* (gray) and *R. castenholzii* (wheat). **g** The residues that coordinate heme 4 are different between *T. tepidum* (gray, accession code 3WMM) and *R. castenholzii* (wheat). The color codes for *R. castenholzii* are the same as Fig. 1

In *tt*RC–LH1, an N-terminal helix of LH1-α occupies the space of B800 in LHβ of *rc*RC–LH, and therefore eliminates the possibility of B800 binding to LH1 at the same position (Fig. 3b). However, in LH2 from *Rhodospirillum molischianum*[9] and LH2 and LH3 from *Rhodopseudomonas acidophila*[35,36], although a short N-terminal helix of LH2-α/LH3-α occupies the space of B800 in LHβ of *rc*RC–LH (Fig. 3c), their B800 molecules can still bind to LH2/LH3 with a different ligation and a distinct orientation, thus spanning a smaller angle onto the membrane in comparison to that of B800 in *rc*RC–LH. Indeed, the LD spectroscopic measurements clearly indicated that the B800 pigments in FAPs are oriented at a large angle with respect to the membrane, in a manner very different from those of purple bacteria[24], which is consistent with our findings. Furthermore, the angles between the transmembrane helices of LHα and LHβ

within a LHαβ heterodimer are all bigger in *rc*RC–LH than in *tt*RC–LH1 (Supplementary Table 6).

We also investigated whether the B880 pigments are arranged in one plane, which might affect the efficiency of energy coupling and transfer. To our surprise, the planarity of B880 pigment arrangement varies among *rc*RC–LH, *tt*RC–LH1, and *rp*RC–LH1 (Fig. 3d), suggesting a possible difference in energy transfer efficiencies among these photosynthetic bacteria. We note the lower planarity in the structure of *rp*RC–LH1 might be due to its limited resolution and map quality[15].

**Architecture of *rc*RC–LH and its quinone shuttling channel.** We further compared the architecture of *rc*RC–LH with that of other core complexes such as *tt*RC–LH1[11] and *rp*RC–LH1[15] by

structural superposition (Fig. 4a). The ring structure of *rc*RC–LH is generally aligned with that of *tt*RC–LH1 and *rp*RC–LH1. However, unlike *tt*RC–LH1, which contains a closed LH1 ring assembled by 16 LH1αβ heterodimers, the LH ring of *rc*RC–LH is assembled by 15 LHαβ heterodimers and has a gap between the 1st and 15th LHαβ heterodimer. The architecture of *rp*RC–LH1 also shows a gap, but the gap locates at the position of the 1st LHαβ (Fig. 4a).

We noted that the TM1 of the L subunit in *rc*RC–LH and the single transmembrane helix of H subunits in both *tt*RC–LH1 and

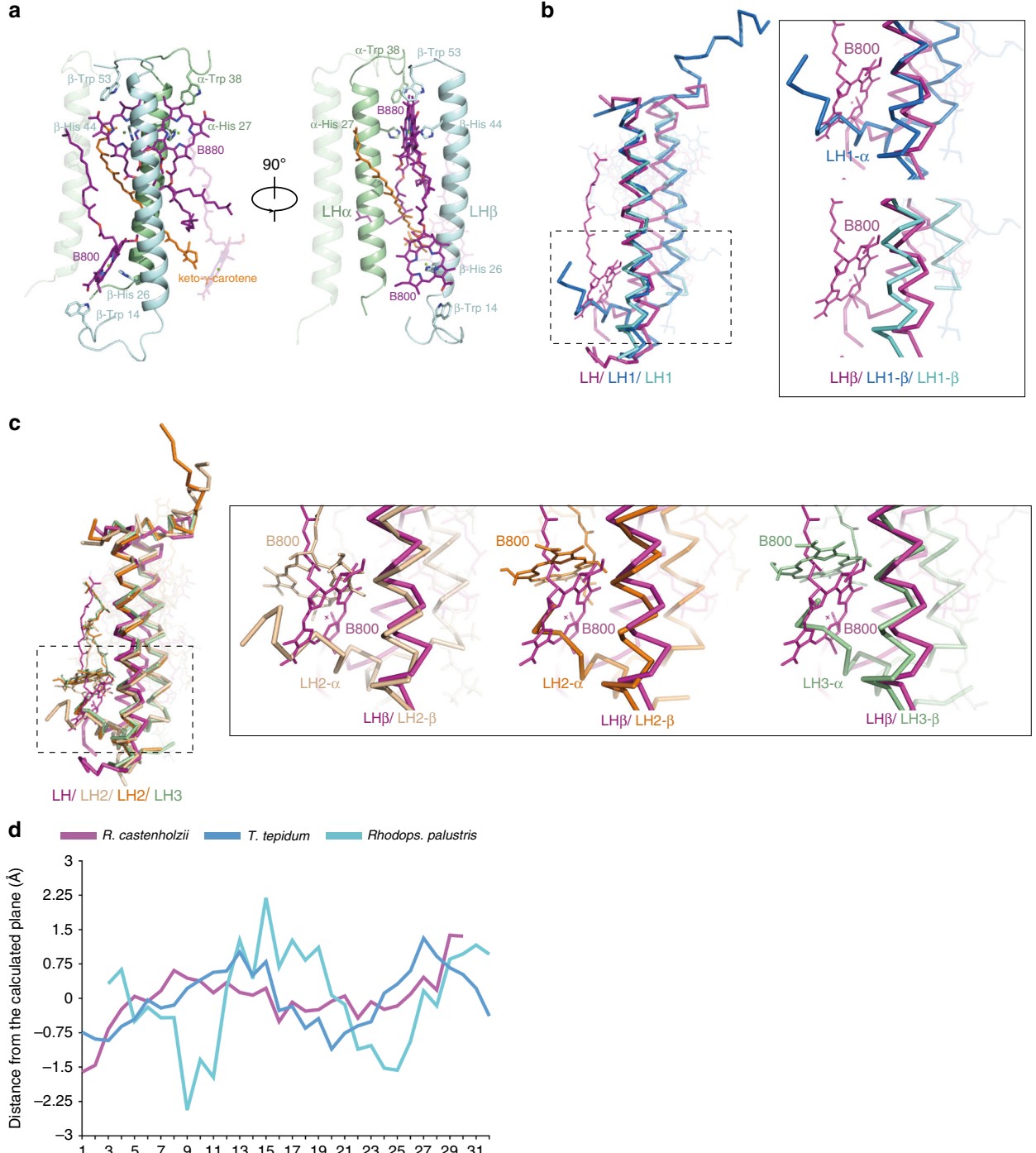

**Fig. 3** Structure of the light-harvesting antenna. **a** Two side views with 90° increment presenting an LHαβ-heterodimer of *R. castenholzii* with cofactors. The neighboring α-apoprotein and B800 are shown with 70% transparency. The BChls (purple), keto-γ-carotene molecules (orange), and their coordinating residues are shown in sticks. **b** An LHαβ-heterodimer of *R. castenholzii* (purple) is compared with the LH1 of *T. tepidum* (blue, accession code 3WMM) and *Rhodops. palustris* (cyan, accession code 1PYH). A zoom-in view of the B800 coordination is shown in the inset. **c** An LHαβ-heterodimer of *R. castenholzii* (purple) is compared with the LH2 of *Rhodospirillum molischianum* (wheat, accession code 1LGH) and LH2 (orange, accession code 1NKZ) and LH3 (pale green, accession code 1IJD) of *Rhodopseudomonas acidophila*. The inset shows a zoom-in view of the B800 coordination. **d** The distances between each B880 pigment and the central plane of B880 pigments ring-array are calculated and plotted to show the planarity of the B880 pigment arrangement for different core complexes, *rc*RC–LH (purple), *tt*RC–LH1 (blue), and *rp*RC–LH1 (cyan)

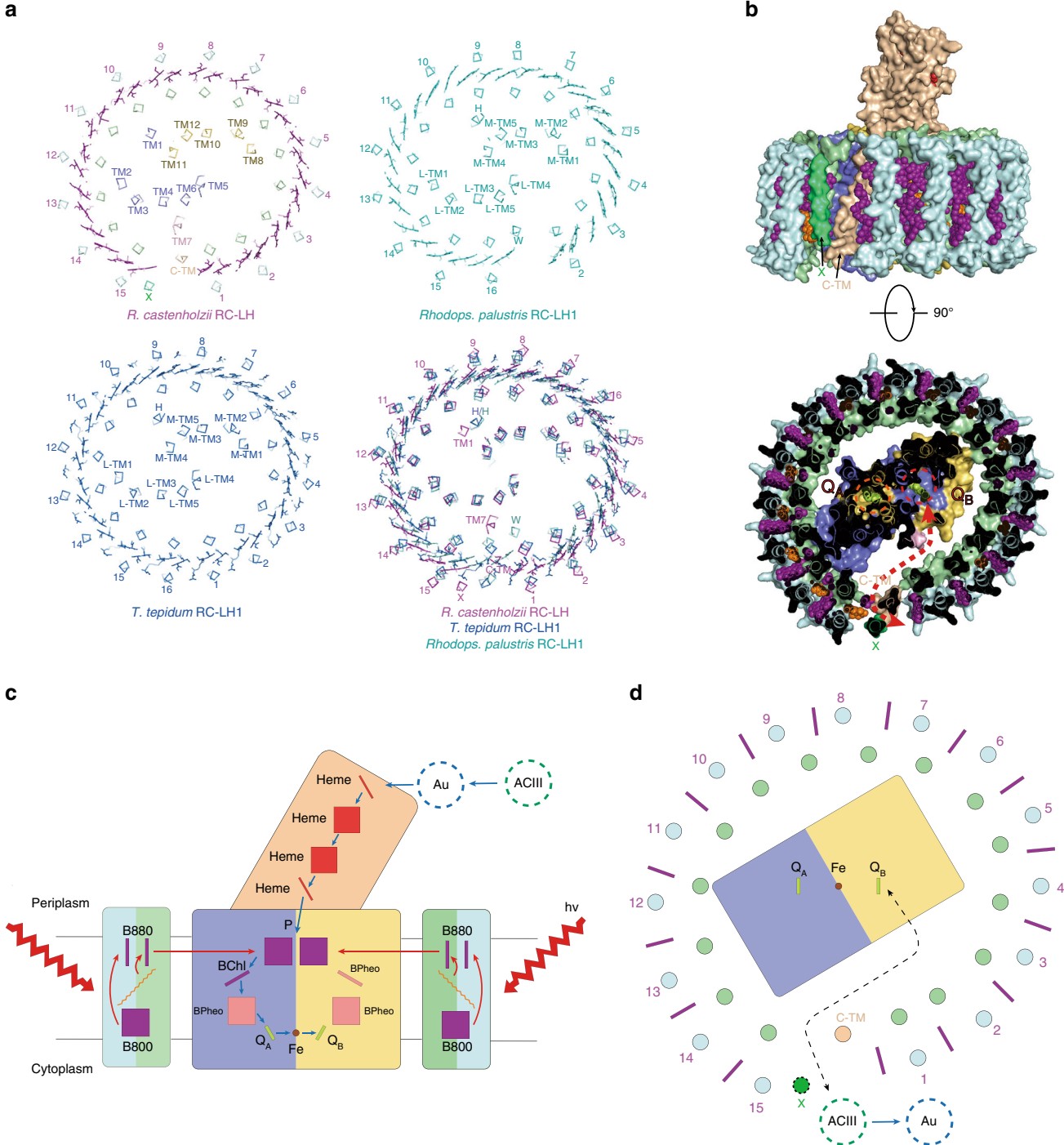

**Fig. 4** Novel architecture of the RC–LH complex of *R. castenholzii* and its quinone shuttling channel. **a** Ribbon representation and comparison of the transmembrane architecture of the core complex from *R. castenholzii* (purple) with that of *T. tepidum* (blue, accession code 3WMM) and *Rhodops. palustris* (cyan, accession code 1PYH). The BChl pigments in LH are shown in sticks. The transmembrane helices of the Cyt *c* subunit, H subunit, protein W, and subunit X are labeled as C-TM, H, W, and X, respectively. **b** The side and bottom-up view of the proposed quinone channel of *rc*RC–LH complex. The BChls and keto-γ-carotene are shown as spheres. The gap between the C-TM and the 15th LHαβ is proposed to be the quinone escape channel. The quinone-binding sites are highlighted by red and orange circles, and the possible quinone shuttling path is shown as red arrows. **c** Schematic model of the energy and electron transfer in *rc*RC–LH complex. The model shows one cross-section that is perpendicular to the membrane. The B800, keto-γ-carotene, and B880 are highly conjugated and the energy from sunlight can be harvested and transferred efficiently among them (red arrows). The energy from the excited B880s also can transfer to the special-pair BChls (P), and facilitate the charge separation. The electron can then transfer to $Q_B$ through BChl, BPheo, $Q_A$, and iron atom sequentially (blue arrows). The $P^+$ receives one electron from heme of RC-attached tetra-heme Cyt *c* and the electron donor of heme is the blue copper protein auracyanin (Au), which is reduced by alternative complex III (ACIII). This diagram was created by Abode Illustrator. **d** The cross-section parallel to the membrane is shown as a schematic model for the quinone transfer. The LH ring barrier possesses one gate between C-TM and the 15th LHαβ for quinone shuttling, which is flanked by subunit X. Fully reduced quinone (hydroquinone) diffuses out of the RC and is replaced by a new quinone. The hydroquinone can transfer electrons to ACIII and then reduce the Au. The color code of all panels is same as Fig. 1

*rp*RC–LH1 occupy a similar position (Fig. 4a), implying a similar role of TM1 for the assembly and stability of *rc*RC–LH as that of H subunit in *tt*RC–LH1 or *rp*RC–LH1. The TM7 of *rc*RC–LH occupies a position that is close to the single transmembrane helix of the W subunit in *rp*RC–LH1 (Fig. 4a), and thus it would take on a similar role for forming the gap in *rc*RC–LH as that of the W subunit in *rp*RC–LH1[15]. Unexpectedly, we observed that the C-TM of Cyt *c* subunit in *rc*RC–LH occupies the position close to the 16th LH1α in the closed ring of *tt*RC–LH1. The C-TM helix partially fills the gap, stabilizes the LH ring and also establishes a connection between RC and LH in the *rc*RC–LH complex (Figs. 1a, b and 4a). Another interesting feature of the *rc*RC–LH architecture, which other core complexes do not have, is the flexible transmembrane helix TMx, it flanks the gap of the LH ring (Figs. 1b and 4a). With these novel architectures, we propose that *rc*RC–LH represents a new type of core complex in anoxygenic phototrophs.

How the reduced quinone penetrates the LH ring and shuttles between the RC and the quinone pool is still controversial. It is highly related with the architecture of the surrounding light-harvesting antenna. For the closed LH1 ring of *tt*RC–LH1, the quinone channel was proposed to be at the interface between each pair of LH1αβ heterodimers on the cytoplasmic side[11], which was also suggested from molecular dynamics simulation studies that quinone can diffuse through the closed LH1 ring with the calculated passage time as ~8 ms, shorter than the turnover rate of the photo-reactive special pair BChl[12]. For the unclosed LH1 ring of *rp*RC–LH1, the gap of the LH1 ring that is interrupted by protein W or Puf X was suggested as the channel for quinone shuttling[14,15], which was further supported by a gene deletion study[37].

However, in the *rc*RC–LH complex, the monomeric B800s block the proposed quinone channels in the closed LH1 ring of *tt*RC–LH1 (Figs. 3b and 4b). Although the LH ring is opened between the 1st and 15th LHαβ heterodimers, the C-TM inserts between these two heterodimers (Figs. 1a, c and 4a), leaving a slit between the C-TM and 15th LHαβ, through which the reduced quinol could pass through to the quinone pool (Fig. 4b). Indeed, C-TM is closer to LHα1 with a distance of 9.8 Å and has a longer distance of 18.4 Å to LHα15. The intra-helical distances of other α-helices within the ring are found varying from 14.1 Å to 15.3 Å in the *rc*RC–LH complex (Supplementary Table 7), which is similar to that of the closed ring in *tt*RC–LH1[11]. Moreover, there is no B800 pigment that exists between C-TM and LHα15 (Fig. 1c, d). Thus, it is conceivable that C-TM inclines to the side of LHα1 and makes a larger distance from LHα15, allowing quinone shuttling (Fig. 4a, b). In addition, the flanking helix TMx could play a role in stabilizing the outer LH ring and might serve as a site to regulate the quinone exchange (Fig. 4b).

## Discussion

*R. castenholzii* acquired a simplified photosynthetic system for efficient solar energy absorption and transformation during evolution. It harvests light through a mosaic LH antenna that combines the topology of LH1 and spectroscopic features of LH2 from purple bacteria. In the current study, we report the cryo-EM structure of the RC–LH core complex from *R. castenholzii* and observe a novel architecture with a gap in the LH ring, the insertion of a Cyt *c* transmembrane helix into the LH ring, and an unusual flexible helix TMx. The RC is assembled by a processed L, M subunit with an extra TM7, and a membrane-bound Cyt *c* subunit.

Based on the architecture of *rc*RC–LH, we tentatively propose a model for its energy and electron transfer mechanism (Fig. 4c, d).

In each LHαβ heterodimer, light energy is absorbed by efficiently coupled pigments (B800, B880, and keto-γ-carotene), and the overall arrangement of LHαβ heterodimers ensures all the excited B880s can transfer energy to the special pair of the RC with approximately the same rate. Once excited, primary charge separation occurs and an electron in the special pair is transferred to the primary electron acceptor BChl in several picoseconds, and is then passed through BPheo, $Q_A$, and iron to $Q_B$. The second primary reaction of the RC fully reduces menaquinone-11 to hydroquinone. The reduced hydroquinone then diffuses from its binding site to the membrane pool through a gap in the LH ring. The hydroquinone is further oxidized by a novel alternative complex (ACIII) found in FAPs that functionally replaces the Cyt $bc_1$ complex of purple bacteria[33], and the electron released during this redox reaction is further transferred to a blue copper protein called auracyanin and finally transferred back to the RC via four hemes bound in the Cyt *c* subunit at the periplasmic side (Fig. 4c). Specifically, the unique C-TM not only associates the Cyt *c* subunit with the RC–LH for rapid electron donation to the special pair, but also, together with the TMx, compensates the opened LH ring to facilitate the hydroquinone transfer.

Overall, our current study reveals the distinctive architecture of the photosystem of an early branching prokaryote, indicates how the energy is transferred between the mosaic LH and the smallest RC, and suggests an interesting quinone exchange model. Notably, identification of the B800-binding sites in the LH provides a structural basis for understanding its function in this unusual energy transfer pathway. In addition, since the L and M subunits in *rc*RC–LH complex are encoded by a fused gene, how these two subunits are processed and assembled into the mature complex, and the assignment of TM7, need further investigation.

## Methods

**Extraction and purification of the *rc*RC–LH complex**. Isolation and purification of the photosynthetic RC–LH complex from photoheterotrophically grown *Roseiflexus castenholzii* cell was carried out by the method as described[25,38] with some modifications. The whole membranes were selectively solubilized by 2% β-DDM at room temperature for 30 min and ultra-centrifuged at 200,000 × *g* for 3 h, the supernatant was collected, taking care to avoid the soft pellet. The reddish-brown supernatant was filtered through a 0.2 μm filter and diluted with Buffer A (0.02% DDM, 50 mM Tris-HCl, pH 8.0) before chromatographic purification. The core complex was isolated by anion exchange chromatography via QSHP5 column (GE Healthcare) and eluted with 200 mM NaCl in the buffer A, further purified by gel filtration on the Superdex 200 16/60 column (GE Healthcare) in buffer B (0.02% DDM, 150 mM NaCl, 50 mM Tris-HCl, pH 8.0). The final 880/280 nm absorption ratio for the core complex was above 1.55. The whole preparation procedure was monitored via the absorption spectrum (250–1000 nm) and SDS-PAGE and blue-native PAGE analysis.

**Electron microscopy**. Three μL aliquots of 3 mg mL$^{-1}$ purified *rc*RC–LH complex in buffer B were placed on glow discharged holey carbon grids (GiG Au R1/1, 300 mesh), and flash frozen in liquid ethane with an FEI Vitrobot IV. Each grid was blotted for 3 s with blotting force level 0 and 100% humidity at 16 °C. Data were collected on a 300 kV FEI Titan Krios electron microscopy with a direct-electron device using FEI Falcon II in integrating mode or FEI Falcon IIIEC (beta version) in counting mode (model C). With Falcon II, a total number of 1858 micrographs were recorded at a nominal magnification of ×59,000 and a pixel size of 1.396 Å, with a dose rate of approximately 24 e$^-$ Å$^{-2}$ s$^{-1}$, and a defocus range between 1.0 and 4.0 μm. Exposure of 2 s was dose fractionated into 32 frames. With Falcon IIIEC, a total number of 2330 micrographs were recorded at a nominal magnification of ×75,000 (yielding a pixel size of 1.10 Å), with a dose rate of approximately 0.8 e$^-$ Å$^{-2}$ s$^{-1}$, and a defocus range between 1.0 and 3.5 μm. Exposure of 50 s yielded 50 frames in total.

**Image processing**. We use Unblur[39] for whole-frame motion correction and exposure weighting, Gctf[40] for estimation of global and local contrast transfer function (CTF) parameters, Gautomatch (developed by Zhang K, MRC Laboratory of Molecular Biology, Cambridge, UK) for automatic particle picking, EMAN2[41] for manual particle picking and initial model generation, and Relion-1.4[42] for all other image processing steps. The templates for automatic particle picking in Gautomatch was 2D class-averaged images of manually picked 1261 particles from 50 micrographs acquired from Falcon II camera. In total, 201,353 (Falcon II)

particles and 522,996 (Falcon IIIEC) were auto-picked from 1858 (Falcon II) and 2330 (Falcon IIIEC) micrographs. For the particles from Falcon II, we used the reference-free 2D classification and reference-based 3D classification to select 89,081 particles for refinement, which yielded a reconstruction in 13.4 Å resolution. For the particles from Falcon IIIEC, the reference-free 2D classification and manual screening were carried out alternately to remove the overlap or bad particles, 323,578 particles were left for further processing. Then, the particles acquired by Falcon IIIEC were subjected to an initial run of 3D classification with four classes, and then 265,123 particles were selected for a first 3D refinement. These particles gave a reconstruction with a resolution of 4.3 Å. After two rounds of 3D classification without performing any alignments, a subset of 148,618 particles was selected for the final refinement. The resolution of the final map was 4.1 Å. Reported resolutions were based on the gold standard FSC = 0.143 criterion. All 3D classifications and refinements were started from a 40 Å low-pass filtered initial model. The final map reconstructed from Falcon IIIEC camera data set used for atomic modeling was corrected for the modulation transfer function of the detector and sharpened by applying an empirically determined B-factor of $-100$ Å$^2$. The values of angular distribution of particles from 3D refinement was visualized by UCSF Chimera[43]. Local resolution variation was estimated with ResMap[44].

**Model building and refinement**. The crystal structure of RC–LH1 from *T. tepidum*[11] (*tt*RC–LH1, accession code 3WMM) was first fitted into the density map in UCSF Chimera[43]. The amino acid sequence alignments were calculated by Clustal Omega[45] and presented by ESPript[46]. The secondary structure prediction was conducted by YASPIN[47]. The conserved residues coordinated the cofactors (BChls and hemes) and the Trp residues in predicted α-helix were assigned. Then, the rest residues were manually built in COOT[48]. Sequence assignment was also guided by the residues having bulky side chains. The cofactors BChl, BPhe, and heme were extracted from the structure of *tt*RC–LH1 and refined in COOT. The menaquinone-11 and keto-γ-carotenes were generated and refined in COOT with restraints from ProDug in CCP4[49–51]. This model was real space refined in PHENIX[52]. The refinement and model statistics are listed in Supplementary Table 3. All structural figures here were generated with UCSF Chimera[53] and PyMOL (www.pymol.org).

**N-terminal protein sequencing**. The purified *rc*RC–LH complex was separated on a 16% Tricine SDS-PAGE gel[54]. The running conditions were 30 V for 1 h followed by 150 V for 5 h. The gel was transferred onto a polyvinylidene difluoride (PVDF) membrane by electrophoresis at 6 V for 18 h at 4 °C. The bands of Cyt *c*, L, and M subunits were excised manually from the membrane, and then were used for N-terminal sequencing by Edman degradation, which was performed using PROCISE491 Sequencer (Applied Biosystems).

**Mass spectrometry**. The protein bands were manually excised from polyacrylamide gels, and were de-stained and dehydrated. Disulfide bonds were reduced with 10 mM DTT for 45 min at 56 °C, and the free sulfhydryl groups were alkylated with 55 mM iodoacetamide for 60 min at room temperature in dark. Afterward, the protein band of LHβ subunit was digested overnight by chymotrypsin at 30 °C, whereas the other samples were digested overnight by trypsin at 37 °C. The reactions were terminated by adding trifluoroacetic acid to a final concentration of 1%.

For MALDI-TOF/TOF mass spectrometry, the samples were desalted using C18 Zip-Tip micro-columns, and were loaded into the instrument in a crystalline matrix of α-cyano-4-hydroxycinnamic acid (CHCA). MALDI-TOF/TOF-MS detection was achieved using an ultrafleXtreme MALDI TOF/TOF mass spectrometer (BRUKER). The data analysis was performed with MS/MS Ions Search of Mascot Server (MATRIX SCIENCE).

Nano-flow liquid chromatography LTQ-Orbitrap mass spectrometric analyses were performed on Easy nLC 1200 system equipped with nanoLC-LTQ-Orbitrap XL mass spectrometers (Thermo, San Jose, CA) at a resolution of 60,000. The raw data were processed by Proteome Discoverer (version 1.4.0.288, Thermo Fisher Scientific). MS2 data were searched with SEQUEST engine against the genomic database of *Roseiflexus castenholzii*.

**Data availability**. Cryo-EM maps and atomic coordinates of *rc*RC–LH have been deposited into Electron Microscopy Data Bank (accession code, EMD-6828) and Protein Data Bank (accession code 5YQ7), respectively. Other data are available from the corresponding authors on reasonable request.

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

## Acknowledgements

We like to thank Prof. Tingyun Kuang from Center of Photosynthesis research, Institute of Botany, Chinese Academy of Science for long-term support on this work. We thank Ming Zhang, Ling Zhang, Gengxin Hu, and Yu Liu from Yueyong Xin's lab, and Xiaoyan Wang and Wanrong Tang from Xiaoling Xu's lab at Hangzhou Normal University for their contributions in biochemical and preliminary structural analyses of the RC–LH complex. We are grateful to Dr. Gang Ji from Center for Biological Imaging (CBI, http://cbi.ibp.ac.cn), Institute of Biophysics (IBP), Chinese Academy of Science (CAS) for his help on cryo-electron microscopy, and Ping Shan and Ruigang Su from Fei Sun's lab in lab management. This work was supported by grants from the Strategic Priority Research Program of Chinese Academy of Sciences (XDB08030202) to F.S., the National Basic Research Program (973 Program) of Ministry of Science and Technology of China (2014CB910700 to F.S. and 2011CBA00904 to Y.Y.X.), and the grants from National Natural Science Foundation of China (31570738 and 31400630) to X.L.X. and Zhejiang Provincial Natural Science Foundation of China (LY14C050002) to X.L.X. The RC–LH complex isolation and purification were performed at the School of Life Science and the structural platform in Institute of Aging Research, School of Medicine at Hangzhou Normal University. The cryo-EM work was performed at CBI, IBP, CAS. All the intensive computations of image processing were performed on the high performance cluster of CBI.

## Author contributions

F.S. and X.L.X. initiated the project and supervised all experiments. Y.Y.X. and Q.Q.W. isolated and purified the RC–LH complex from R.castenholzii. W.Q.N. performed biochemical and preliminary structural analyses of the RC–LH complex. Y.S. performed single-particle cryo-EM grid preparation, data collection, all model building and refinement. T.X.N. and Y.S. performed images processing. X.J.H. and W.D. built data collection pipeline based on Falcon IIIEC camera (beta version). L.Y. analyzed the mass spectrometry data during revision of the manuscript. Y.Y.X., Y.S., R.E.B., X.L.X., and F.S. analyzed the data and wrote the manuscript.

## Additional information

**Competing interests:** The authors declare no competing interests.

