## [Peer Review File · Nature Communications]

Reviewers' comments:

Reviewer #1 (Remarks to the Author):

This is a fascinating new structure of a completely unusual LH/RC 'core' complex. This is an important piece of work but is currently rather poorly written. It needs correction by an English speaking person(Blankenship?). Basically the structure is well described and contains a raft of features not seen before. The authors should however be more circumspect in suggesting 'quinone gating' as there is no evidence to show there is indeed an active gating mechanism. Calling the mystery peptide LHgamma is too pejorative. A more neutral name would be better without definite proof of the origin of this peptide. In comparing the organisation of the Bchl macro-cycles in the different published core complex structures especially in the RP structure the positions should not be overinterpreted. The resolution of that structure did not allow accurate positioning of these rings and so comments on that should be toned down.

Reviewer #2 (Remarks to the Author):

In the manuscript "Cryo-EM structure of the RC-LH core complex from an early branching photosynthetic prokaryote" Xin et al. describe the structure of the photosynthetic core complex of *R. castenholzii*. The authors build the structure of the reaction center PufLM and cytochrome C and the light harvesting antenna ring (LH ring) composed of PufA and PufB into a 4.3 Å cryo-EM density. The structure reveals an elliptical shape of the LH ring with 15 well-resolved subunits and "gap" area where the authors locate a transmembrane helix of *R. castenholzii* cytochrome C. Similar structure of the bacterial photosynthetic core complexes have been solved by crystallography (*R. palustris* - Roszak et al; *T. tepidum* - Niwa et al) revealing the overall architecture of the complex. The structure of the core subunits is conserved and the *T. tepidum* structure shows as similarly bound cytochrome C. The transmembrane helix of the H subunit of in the crystal structures is replaced by TM1 of the L subunit in *R. palustris*. The structure of the LH dimer is conserved but the B800 pigments are oriented in a different angle with respect to the membrane. The arrangement of the LH dimers in the present structure is more similar to the *R. palustris* structure in which the LH ring is also not closed and the PufX protein is found in the resulting gap. It has been suggested for the *R. palustris* structure that the gap allows access of quinone to transfer electrons to cytochrome C. In the gap region the authors locate two additional transmembrane helices that are not assigned, in addition to the transmembrane helix attributed to cytochrome C. The authors speculate that these helices are involved in regulation of quinone transfer, however without an identification of the protein(s) the additional insights provided by the structure are limited.

The single particle reconstruction workflow is well documented showing a raw micrograph of with particles of sufficient quality and average images that match the structure. The local resolution map of the structure shows higher resolution than average in the core and less well resolved outer domains similarly to other macromolecular complexes. The core of the structure is probably resolved at <4 Å and could thus be built. Unfortunately, the authors did not employ 3D classification (In the method section the authors state "reference based 3D classification did not work"). In most single particle reconstruction workflows a 3D classification step is essential to achieve sorting into more homogeneous particle populations resulting in higher resolution reconstructions. The 3D classification can also be used to sort for ligand occupancy or conformational changes in part of the structure (e.g. in Relion: Bai XC, Rajendra E, Yang G, Shi Y, Scheres SH. Sampling the conformational space of the catalytic subunit of human γ -secretase. *Elife*. 2015). Applying 3D classification techniques could improve the local resolution and occupancy of the unassigned transmembrane segment LHgamma and might help to assign the transmembrane segments. The number of particles 256903 would allow sorting into more homogenous subsets as only approx. 25000 - 50000 particles should be required to reach 4 Å resolution. Furthermore, the authors could analyze the sample by mass spectrometry to identify potential additional protein components of the complex.

The refinement statistics of the built part of the structure show a comparatively large number of Ramachandran outliers and residues in the Ramachandran allowed region. Thus, the structure should be checked carefully for building errors.

In conclusion, the structure reveals the overall architecture of the photosynthetic core complex of *R. castenholzii*, however the single particle reconstruction process could be significantly improved to obtain a better-resolved density. This would allow for more detailed insights into the functionally discussed gap region of the LH ring.

Minor points:

#The overlay of 4 structures in Figure 3c is confusing. A pairwise comparison might be clearer.

#In Figure S3 the density of the transmembrane helix in cyt C and the density for TM1 of the L subunit should be shown in with the modelled side chains.

#In the supplement, the density of LHgamma should be shown at different density threshold levels together with surrounding LH dimers to help the reader evaluate the quality of the density for LHgamma.

Response to Reviewers

We would like to express our appreciation to both reviewers for their positive comments upon our work and their invaluable suggestions to improve the quality of the manuscript.

During the revision process, we carried out further image processing by utilizing different software packages and performing decent 3D classification, which enabled us to improve the resolution from 4.3 to 4.1 angstrom according to the gold standard FSC0.143 criteria. With a better quality of the density map, we further refined the atomic model of the complex. Therefore, our discoveries about the transmembrane helix of Cyt *c*, the processed TM7 and the flexible transmembrane helix at the gap of LH ring have been further solidified.

Besides, we also carried out more experiments including high-resolution Tricine SDS-PAGE, blue native PAGE and mass spectrometry (see Supplementary Fig. 1 and Supplementary Data 1 and 2). These new data not only further verified the existences of known subunits (α , β , L, M, and Cyt *c*) in the complex, but also identified a new subunit X of the complex. Sequence analysis suggested this subunit X comprises a single transmembrane helix, which coincidentally matches the discovery of the flexible transmembrane helix at the gap of the LH ring. Thus the flexible transmembrane helix that might be the part of the subunit X has been renamed as TMx in the revision (it was called LHgamma in the previous submitted version).

In addition, we also repeated the N-terminal sequencing experiments to improve the quality of the data (see Supplementary Fig. 1). With these confirmed data, the statement about the origin of TM7, '*Thus, TM7 could be either an unidentified new gene product or a proteolytic fragment of the pufLM gene product that is cleaved twice during processing*' (Line 246-248), has been becoming more circumspect.

The manuscript has been carefully revised according to the reviewers' comments and the discoveries have been stated in a more rigorous way. The language has been also polished by Prof. Robert E. Blankenship further. As a result, we are confident that the current revision is in good shape and ready for further consideration.

The reviewers' comments/suggestions are responded to in detail below.

Reviewer #1:

[Comments]

This is a fascinating new structure of a completely unusual LH/RC 'core' complex. This is an important piece of work but is currently rather poorly written. It needs correction by an English speaking person (Blankenship?). Basically the structure is well described and contains a raft of features not seen before. The authors should however be more circumspect in suggesting 'quinone gating' as there is no evidence to show there is indeed an active gating mechanism. Calling the mystery peptide LHgama is too pejorative. A more neutral name would be better without definite proof of the origin of this peptide. In comparing the organisation of the Bchl macro-cycles in the different published core complex structures especially in the RP structure the positions should not be overinterpreted. The resolution of that structure did not allow accurate positioning of these rings and so comments on that should be toned down.

[Responses]

1. This is an important piece of work but is currently rather poorly written. It needs correction by an English speaking person (Blankenship?).

We would like to thank this reviewer for his/her positive comment of our work. The manuscript has been revised carefully with new data and the language has been thoroughly polished by the coauthor, Prof. Robert E. Blankenship.

2. The authors should however be more circumspect in suggesting 'quinone gating' as there is no evidence to show there is indeed an active gating mechanism.

We appreciate the rigorous comment of this reviewer. Yes, for the newly identified flexible transmembrane helix (named TMx in the revision) at the LH ring gap, the statement of its role has been modified as follows.

'In addition, the flanking helix TMx could play a role in stabilizing the outer LH ring and might serve as a site to regulate the quinone exchange.' (Line 374-375).

The concept of 'quinone gating' has been deleted in the revision. And the abstract has been revised as follows.

'..... A unique N-terminal transmembrane helix of cytochrome c inserts into the LH ring, not only yielding a tight bound cytochrome c for rapid electron transfer but also opening a slit of the LH ring, which is further flanked by a transmembrane helix from a newly discovered subunit X. These novel structural features suggest a unique quinone exchange model of prokaryotic photosynthetic machinery.'

3. Calling the mystery peptide LHgama is too pejorative. A more neutral name would be better without definite proof of the origin of this peptide.

We would like to thank the reviewer for this rigorous comment . Yes, we agree that a more neutral name would be more appropriate. During the revision process, we performed peptide mass fingerprinting analysis of the core complex. A hypothetical subunit X containing 63 residues was identified (see Supplementary Fig. 1A and Supplementary Data), which coincidentally matches the discovery of the flexible transmembrane helix at the LH ring gap. Thus the mystery peptide LHgama has been renamed as TMx, which might be part of the subunit X. The origin of this peptide needs to be further verified in future investigations.

4. In comparing the organisation of the Bchl macro-cycles in the different published core complex structures especially in the RP structure the positions should not be overinterpreted. The resolution of that structure did not allow accurate positioning of these rings and so comments on that should be toned down.

We would like to thank the reviewer for these careful comments. Yes, we further investigated the maps of these three complexes and found the map qualities of our structure and *ttRC-LH1* are good enough to locate the positions of Bchls with a high precision. However, for the structure of *rpRC-LH1*, the precision of the Bchls' locations are limited by the quality of the map. Thus, the statement about the organization of the Bchl macro-cycles in the manuscript has been revised as follows.

'To our surprise, the planarity of B880 pigment arrangement varies among *rcRC-LH*, *ttRC-LH1* and *rpRC-LH1* (**Fig. 3D**), suggesting a possible difference in energy transfer efficiencies among these photosynthetic bacteria. To be

noted that, the lower planarity in the structure of *rpRC-LH1* might be due to its limited resolution and map quality.' (Line 321-326).

Reviewer #2:

[Comments]

In the manuscript "Cryo-EM structure of the RC-LH core complex from an early branching photosynthetic prokaryote" Xin et al. describe the structure of the photosynthetic core complex of *R. castenholzii*. The authors build the structure of the reaction center PufLM and cytochrome C and the light harvesting antenna ring (LH ring) composed of PufA and PufB into a 4.3 Å cryo-EM density. The structure reveals an elliptical shape of the LH ring with 15 well-resolved subunits and "gap" area where the authors locate a transmembrane helix of *R. castenholzii* cytochrome C.

Similar structure of the bacterial photosynthetic core complexes have been solved by crystallography (*R. palustris* - Roszak et al; *T. tepidum* - Niwa et al) revealing the overall architecture of the complex. The structure of the core subunits is conserved and the *T. tepidum* structure shows as similarly bound cytochrome C. The transmembrane helix of the H subunit of in the crystal structures is replaced by TM1 of the L subunit in *R. palustris*. The structure of the LH dimer is conserved but the B800 pigments are oriented in a different angle with respect to the membrane. The arrangement of the LH dimers in the present structure is more similar to the *R. palustris* structure in which the LH ring is also not closed and the PufX protein is found in the resulting gap. It has been suggested for the *R. palustris* structure that the gap allows access of quinone to transfer electrons to cytochrome C. In the gap region the authors locate two additional transmembrane helices that are not assigned, in addition to the transmembrane helix attributed to cytochrome C. The authors speculate that these helices are involved in regulation of quinone transfer, however without an identification of the protein(s) the additional insights provided by the structure are limited.

The single particle reconstruction workflow is well documented showing a raw micrograph of with particles of sufficient quality and average images that match the structure. The local resolution map of the structure shows higher

resolution than average in the core and less well resolved outer domains similarly to other macromolecular complexes. The core of the structure is probably resolved at <4 Å and could thus be built. Unfortunately, the authors did not employ 3D classification (In the method section the authors state “reference based 3D classification did not work”). In most single particle reconstruction workflows a 3D classification step is essential to achieve sorting into more homogeneous particle populations resulting in higher resolution reconstructions. The 3D classification can also be used to sort for ligand occupancy or conformational changes in part of the structure (e.g. in Relion: Bai XC, Rajendra E, Yang G, Shi Y, Scheres SH. Sampling the conformational space of the catalytic subunit of human γ -secretase. *Elife*. 2015). Applying 3D classification techniques could improve the local resolution and occupancy of the unassigned transmembrane segment LH γ and might help to assign the transmembrane segments. The number of particles 256903 would allow sorting into more homogeneous subsets as only approx. 25000 - 50000 particles should be required to reach 4 Å resolution. Furthermore, the authors could analyze the sample by mass spectrometry to identify potential additional protein components of the complex.

The refinement statistics of the built part of the structure show a comparatively large number of Ramachandran outliers and residues in the Ramachandran allowed region. Thus, the structure should be checked carefully for building errors.

In conclusion, the structure reveals the overall architecture of the photosynthetic core complex of *R. castenholzii*, however the single particle reconstruction process could be significantly improved to obtain a better-resolved density. This would allow for more detailed insights into the functionally discussed gap region of the LH ring.

Minor points:

#The overlay of 4 structures in Figure 3c is confusing. A pairwise comparison might be clearer.

#In Figure S3 the density of the transmembrane helix in cyt C and the density for TM1 of the L subunit should be shown in with the modelled side chains.

#In the supplement, the density of LHgamma should be shown at different density thresholds levels together with surrounding LH dimers to help the reader evaluate the quality of the density for LHgamma.

[Responses]

1. In the gap region the authors locate two additional transmembrane helices that are not assigned, in addition to the transmembrane helix attributed to cytochrome C. The authors speculate that these helices are involved in regulation of quinone transfer, however without an identification of the protein(s) the additional insights provided by the structure are limited.

We would like to thank the reviewer for this rigorous comment. As is described above, we have performed more experiments to verify those newly identified transmembrane helices. And the statement of their roles have been also revised in a rigorous way.

We carried on more experiments including high-resolution Tricine SDS-PAGE, blue native PAGE and mass spectrometry (see Supplementary Fig. 1 and Supplementary Data 1 and 2). These new data not only further verified the existences of known subunits (α , β , L, M, and Cyt c) in the complex, but also identified a new subunit X of the complex. Sequence analysis suggested this subunit X comprises a single transmembrane helix, which coincidentally matches the discovery of the flexible transmembrane helix at the LH ring gap. Thus the flexible transmembrane helix might be the part of this subunit X, and we assigned this flanking transmembrane helix as TMx, to discriminate it from the TM helices of LH α / β subunit. Whatever, the origin of this TMx needs to be further verified in the future investigations.

As we responded to Reviewer #1, the statement of the role of TMx has been revised as

'In addition, the flanking helix TMx could play a role in stabilizing the outer LH ring and might serve as a site to regulate the quinone exchange.' (Line 374-375).

The concept of 'quinone gating' has been deleted. And the abstract has been revised as follows.

'..... A unique N-terminal transmembrane helix of cytochrome c inserts into the LH ring, not only yielding a tight bound cytochrome c for rapid electron transfer but also opening a slit of the LH ring, which is further flanked by a transmembrane helix from a newly discovered subunit X. These novel structural features suggest a unique quinone exchange model of prokaryotic photosynthetic machinery.'

2. The core of the structure is probably resolved at <4 Å and could thus be built. Unfortunately, the authors did not employ 3D classification (In the method section the authors state "reference based 3D classification did not work"). In most single particle reconstruction workflows a 3D classification step is essential to achieve sorting into more homogeneous particle populations resulting in higher resolution reconstructions. The 3D classification can also be used to sort for ligand occupancy or conformational changes in part of the structure (e.g. in Relion: Bai XC, Rajendra E, Yang G, Shi Y, Scheres SH. Sampling the conformational space of the catalytic subunit of human γ -secretase. *Elife*. 2015). Applying 3D classification techniques could improve the local resolution and occupancy of the unassigned transmembrane segment LHgamma and might help to assign the transmembrane segments. The number of particles 256903 would allow sorting into more homogenous subsets as only approx. 25000 - 50000 particles should be required to reach 4 Å resolution.

We would like to thank this reviewer for his/her expertise on image processing.

During the revision process, we carried out further image processing that includes dose-weighting using Unblur, local CTF estimation using Gctf and decent 3D classification using Relion. As expected from this reviewer, applying further 3D classification techniques did improve the resolution of the final reconstruction from 4.3 to 4.1 Å according to the gold standard FSC0.143 criteria (see Supplementary Fig. 2 and 4). With a better quality of the density map, we could further refine the model of the complex.

However, due to the limited resolution, the side chains of TMx and TM7 could still not be assigned, which, as we believe, is due to the quality of the specimen and the flexible nature of the complex.

3. Furthermore, the authors could analyze the sample by mass spectrometry to identify potential additional protein components of the complex.

As the reviewer suggested, we performed mass spectrometry and did identify a new hypothetical subunit X with 38.1% sequence coverage and appropriate molecular weight (see Supplementary Fig. 1 and Supplementary Data 1 and 2). Please also see our response #1 to this reviewer for more information.

4. The refinement statistics of the built part of the structure show a comparatively large number of Ramachandran outliers and residues in the Ramachandran allowed region. Thus, the structure should be checked carefully for building errors.

We have refined the model based on the improved density map, and the final refined model shows fewer Ramachandran outliers (see Supplementary Table 1).

5. The overlay of 4 structures in Figure 3c is confusing. A pairwise comparison might clearer.

We have updated Fig. 3B and 3C with pairwise comparison of the LH α/β subunit from different species.

6. In Figure S3 the density of the transmembrane helix in cyt C and the density for TM1 of the L subunit should be shown in with the modelled side chains.

The density of TM1 of L subunit, and the C-TM of Cyt c subunit with modeled side chains have been shown in the revised Supplementary Fig. 5B and 5F.

7. In the supplement, the density of LHgamma should be shown at different density thresholds levels together with surrounding LH dimers to help the reader evaluate the quality of the density for LHgamma.

Thanks a lot for this good suggestion. The density of TMx has been shown with different thresholds in Supplementary Fig. 6. In addition, we also made

Supplementary Movies 1 and 2 to show the quality of the map and the model building.

REVIEWERS' COMMENTS:

Reviewer #2 (Remarks to the Author):

The authors addressed the issues raised by the reviewer and the manuscript is now suitable for publication. The additional mass spectrometry data provided corroborate the analysis of the authors regarding the density for the additional transmembrane helix TMx. The additional 3D classification steps for 3D structure calculation improved the maps and the model geometry has been refined. The positioning of structural elements, TM7, in the less well-resolved parts of the map is appropriately discussed.